# Combined deletions of *IHH* and *NHEJ1* cause chondrodystrophy and embryonic lethality in the Creeper chicken

Keiji Kinoshita[1,6,7], Takayuki Suzuki[1,2,7], Manabu Koike[3,7], Chizuko Nishida[4], Aki Koike[3], Mitsuo Nunome[1], Takeo Uemura[1], Kenji Ichiyanagi[5] & Yoichi Matsuda[1,2 ✉]

The Creeper (*Cp*) chicken is characterized by chondrodystrophy in *Cp/+* heterozygotes and embryonic lethality in *Cp/Cp* homozygotes. However, the genes underlying the phenotypes have not been fully known. Here, we show that a 25 kb deletion on chromosome 7, which contains the Indian hedgehog (*IHH*) and non-homologous end-joining factor 1 (*NHEJ1*) genes, is responsible for the *Cp* trait in Japanese bantam chickens. *IHH* is essential for chondrocyte maturation and is downregulated in the *Cp/+* embryos and completely lost in the *Cp/Cp* embryos. This indicates that chondrodystrophy is caused by the loss of *IHH* and that chondrocyte maturation is delayed in *Cp/+* heterozygotes. The *Cp/Cp* homozygotes exhibit impaired DNA double-strand break (DSB) repair due to the loss of *NHEJ1*, resulting in DSB accumulation in the vascular and nervous systems, which leads to apoptosis and early embryonic death.

[1] Avian Bioscience Research Center, Graduate School of Bioagricultural Sciences, Nagoya University, Chikusa-ku, Nagoya, Aichi 464-8601, Japan. [2] Laboratory of Avian Bioscience, Department of Animal Sciences, Graduate School of Bioagricultural Sciences, Nagoya University, Chikusa-ku, Nagoya, Aichi 464-8601, Japan. [3] National Institute of Radiological Sciences, National Institutes for Quantum and Radiological Science and Technology, Inage-ku, Chiba 263-8555, Japan. [4] Department of Natural History Sciences, Faculty of Science, Hokkaido University, Kita-ku, Sapporo, Hokkaido 060-0808, Japan. [5] Laboratory of Genome and Epigenome Dynamics, Department of Animal Sciences, Graduate School of Bioagricultural Sciences, Nagoya University, Chikusa-ku, Nagoya, Aichi 464-8601, Japan. [6] Present address: State Key Laboratory for Conservation and Utilization of Bio-Resources in Yunnan, Yunnan Agricultural University, Kunming 650201, China. [7] These authors contributed equally: Keiji Kinoshita, Takayuki Suzuki, Manabu Koike. ✉email: yoimatsu@agr.nagoya-u.ac.jp

The Creeper (Cp) trait of the chicken is characterized by autosomal dominant inheritance of congenital shortness of extremities caused by chondrodystrophy in Cp/+ heterozygotes; however, the Creeper allele is lethal at the embryonic stage in Cp/Cp homozygotes[1–5]. The Cp locus is closely linked to the MNR2 (MNR2 homeodomain protein) gene on chromosome 7, which is responsible for the Rose-comb trait[4,6–9]. Recently, the causative gene at the Cp locus was reported to be the Indian hedgehog (IHH) gene in the Chinese Xingyi bantam chicken based on whole-genome sequencing analysis[10], which suggested that the decreased level of IHH expression during cartilage development caused chondrodystrophy in Cp/+ heterozygotes and the loss of IHH expression was associated with embryonic death of Cp/Cp homozygotes. Recently, to characterize the Creeper phenotype precisely, we established a congenic strain of the Cp allele, GSP/Cp, which expresses a Cp allele introduced into the Fayoumi inbred GSP (group-specific antigen positive for avian leucosis virus) strain from the Japanese bantam (JB) chicken. It is notable that most of the Cp/Cp homozygous embryos in the GSP/Cp strain died during the early developmental stage around embryonic day 3 (E3) and exhibited multiple morphogenetic abnormalities. This early embryonic lethality before osteogenesis cannot be fully explained by the deletion of IHH, the gene, which is responsible for proliferation and differentiation of the chondrocyrtes[11–16].

In this paper, we show that the causative mutation of the Cp allele is a 25 kb deletion containing the IHH and non-homologous end-joining factor 1 (NHEJ1) genes. Decreased IHH expression causes shortened limbs and smaller body size in Cp/+ heterozygotes, and early embryonic death in Cp/Cp homozygotes is mainly caused by impaired DNA double-strand break (DSB) repair due to the loss of NHEJ1. This occurs concomitantly with the suppression of cell growth.

## Results

**Morphologies of the Cp embryos.** Both the JB and congenic GSP/Cp strains exhibited the Cp trait. The GSP/Cp strain expresses a Cp allele that was introduced from the JB strain into a highly inbred strain derived from the Fayoumi breed, GSP, whose average heterozygosity is <0.01 for >50 microsatellite DNA markers[17] (Supplementary Figs. 1–4 and Supplementary Table 1). The homozygous Cp/Cp embryos in the JB strain had a higher rate of survival during the late stages of development (Fig. 1a); however, almost all the Cp/Cp embryos in the GSP/Cp strain had multiple morphogenetic abnormalities during the early developmental stage around E3, which included the hypoplasia of the brain, especially smaller size of mesencephalon, telencephalon, and diencephalon, and smaller size of heart and the loss of extraembryonic blood vessel formation (Fig. 1b and Supplementary Tables 2 and 3).

**Chondrocyte proliferation and differentiation.** We found that some of the Cp/Cp embryos could survive until E15 in the JB strain. In skeletal staining of the E14 embryos of the JB strain, the Alizarin red-stained regions in the limbs of the Cp/+ embryos were smaller than those observed in the wild-type (+/+) embryos; however, the skeletal pattern was normal (Fig. 1a). This suggests that the endochondral ossification process is down-regulated in Cp/+ heterozygotes, resulting in shorter bones. The Cp/Cp embryos were relatively much smaller than the wild-type and Cp/+ embryos and had no Alizarin red-stained regions in the cartilage. Elongation of both forelimbs and hindlimbs was severely defected, whereas the Alcian blue-stained cartilage bones still existed. Metatarsal-phalangeal joints and phalangeal joints were also deformed in the autopod. These results indicate that

chondrocyte proliferation and subsequent differentiation are completely inhibited in Cp/Cp homozygotes, resulting in shorter limbs and smaller sized embryos.

**Chromosome mapping of the candidate region.** Chromosome mapping with the SNP array revealed 127,180 SNPs with minor allele frequencies <0.15 in the GSP strain. The absolute relative allelic frequency (RAF) difference (absRAFdif) was calculated between the wild-type (+/+) GSP and heterozygous (Cp/+) GSP/Cp chickens, and the peaks with high allelic frequency differences were localized to 18 chromosomes (Supplementary Fig. 5 and Supplementary Table 4). Of the 18 peaks, the highest median value was observed in the 6.1 Mb region at nucleotide positions 16,465,727 to 22,536,077 on chromosome 7 in the chicken reference genome Gallus_gallus-4.0 (GCA_000002315.2) (17032895–23162069 in Gallus_gallus-5.0, GCA_000002315.3).

For linkage mapping of the Cp locus, we first roughly mapped the Cp locus with 16 SNP markers and three microsatellite DNA markers located at nucleotide position 2.52 − 35.19 Mb on chromosome 7 between CHR7_#5 and CHR7_#13 using 95 F$_2$ progeny of the JB and GSP strains (Supplementary Tables 5 and 6). Next, nine SNP markers (CHR7_#2, #3, #4, #14, #15, #16, #17, #18, and #19) and three microsatellite DNA markers (MCW0133, MCW183, and ADL180) within a 15.711 − 26.581 Mb region were used for fine mapping with 175 F$_2$ progeny that also included the 95 individuals used for rough mapping (Supplementary Table 6). Fine mapping allowed us to narrow the candidate region of the Cp locus to approximately 90 kb, between CHR7_#16 and CHR7_#3, at nucleotide position 21,726,140–21,815,955 in the reference genome (Gallus_gallus-4.0).

**Identification of the causative mutation.** A potential genomic deletion was found at nucleotide position 21,786,125–21,810,639 in the 90 kb region by SNP genotyping of the GSP (+/+), GSP/Cp (Cp/+), and JB (Cp/+) chickens and a JB (Cp/Cp) embryo with 119 probes of the SNP array (Supplementary Data 1). To confirm the presence of the genomic deletion in this 25 kb region, we designed a pair of PCR primers in the fifth intron and sixth exon of the Indian hedgehog (IHH) gene (IHH_F: 5′–TCTGTGGT GCTGTCTCATGACC–3′ and IHH_R: 5′–ACTTGACGGAGCA GTGGATGTG–3′) and tested the GSP (+/+), GSP/Cp (Cp/+), JB (Cp/+), and JB (Cp/Cp) samples. The 961 bp fragment was not amplified in the Cp/Cp homozygous embryo (Supplementary Fig. 6), indicating that the IHH amplicon is located in the deleted region.

To identify the breakpoints and the size of the deletion, we designed 11 primer sets (#A–#K) at 4–6 kb intervals within the 56 kb region at 22.395–22.451 Mb in the reference genome (Gallus_gallus-5.0; Supplementary Fig. 7 and Supplementary Table 7). PCR products were obtained for GSP (+/+), GSP/Cp (Cp/+), JB (Cp/+), and JB (Cp/Cp) using the primer sets #A – #D, #J, and #K. No PCR products were amplified from the Cp/Cp homozygote by primer sets #E– #I. This indicates that the 5′ and 3′ breakpoints were located between the #D and #E primers and between the #I and #J primers, respectively. The distance between primers #D and #J was 35 kb in the wild type; however, a 9.9 kb PCR product was amplified from Cp/+ heterozygotes and the Cp/Cp homozygote by LA-PCR using primers #D_fd and #J_rv (Supplementary Fig. 8). To narrow down the deleted region, we designed four forward primers (#DE_fd1–#DE_fd4) in the #D–#E region, including the 5′ breakpoint, and five forward primers (#IJ_fd1–#IJ_fd5) and ten reverse primers (#IJ_rv1–#IJ_rv5, #IJ_delrv1–#IJ_delrv5) in the #I–#J region, including the 3′ breakpoint (Supplementary Fig. 7 and Supplementary Table 8). This series of primers allows us to test if PCR products could be

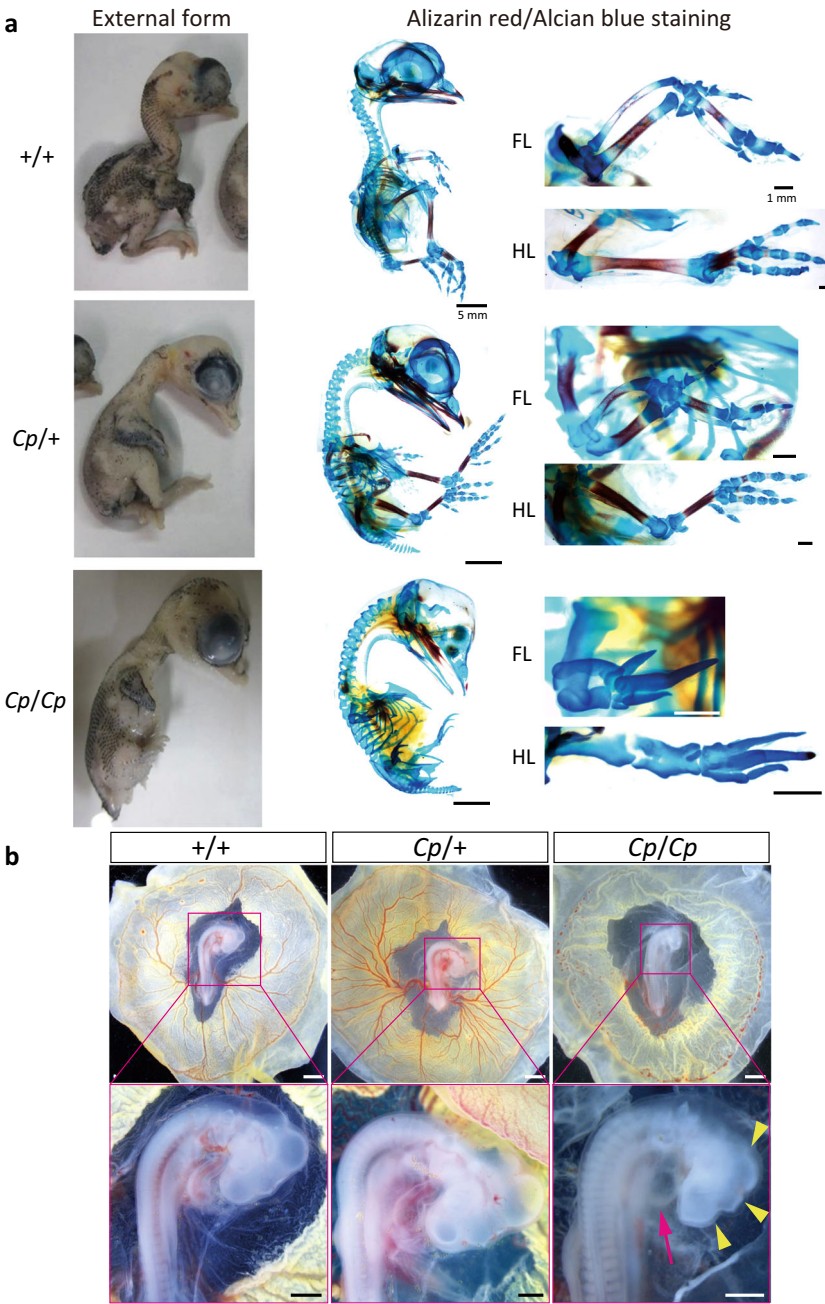

**Fig. 1 Creeper phenotypes of the E14 embryos from the JB strain and the E3 embryos from the GSP/Cp strain. a** For the E14 embryos (JB strain), the heterozygous (*Cp*/+) embryo has short forelimbs and hindlimbs, compared to the wild-type (+/+) embryo. The phenotype of the homozygous (*Cp*/*Cp*) embryo is similar to the embryo with phocomelia syndrome. The lateral-view images of the Alizarin red/Alcian blue-stained embryos demonstrates that the bones and cartilage are visualized with Alizarin red and Alcian blue, respectively. Bar, 5 mm. High-magnification dorsal views of the forelimb (FL) and hindlimb (HL) show that the limbs are oriented with the anterior side up and the posterior side down. The digits are located on the distal side of the limb. Bar, 1 mm. **b** The morphologies of the +/+, *Cp*/+, and *Cp*/*Cp* embryos at E3 from the GSP/Cp strain. The *Cp*/*Cp* embryos displayed hypoplasia of the blood vessels in the yolk sac, malformation, and slow growth of the brain (yellow arrowhead) and heart tube (magenta arrow). The blood cells were not circulated in the *Cp*/*Cp* embryos, and the heart beat slowly. Bar, 1 mm (upper panels), 500 μm (lower panels).

obtained from different genotypes (+/+, *Cp*/+, *Cp*/*Cp*). A 1261 bp PCR product was obtained with the primer set #DE_fd3 and #IJ_delrv5 in the *Cp*/+ heterozygotes and *Cp*/*Cp* homozygote, but not from the wild type (+/+) (Supplementary Figs. 9 and 10). This 1261 bp product contained a 578 bp inverted sequence at position 22,426,651–22,427,228 in the reference genome (Gallus_gallus-5.0), which remained between the deleted regions at position 22,413,168–22,426,650 (13,483 bp) and 22,427,229–22,438,789 (11,561 bp) at the *Cp* locus (Fig. 2; and Supplementary

Figs. 7, 9, and 10a) (accession no. LC332495 is the 1031 bp sequence that includes the 578 bp inverted sequence in the deleted region). This differed greatly from the 11,896 bp deletion in Chinese Xingyi bantam chickens containing only the *IHH* gene (Chr 7: 21,798,705–21,810,600 in Gallus_gallus-4.0)[10]. These deletions were confirmed by LA-PCR using primers DE_fd3 and Invwt_rv that were designed across the 13,483 bp deletion and primers Invwt_fd and IJ_delrv5 that were designed across the 11,561 bp deletion (Supplementary Table 8 and Supplementary

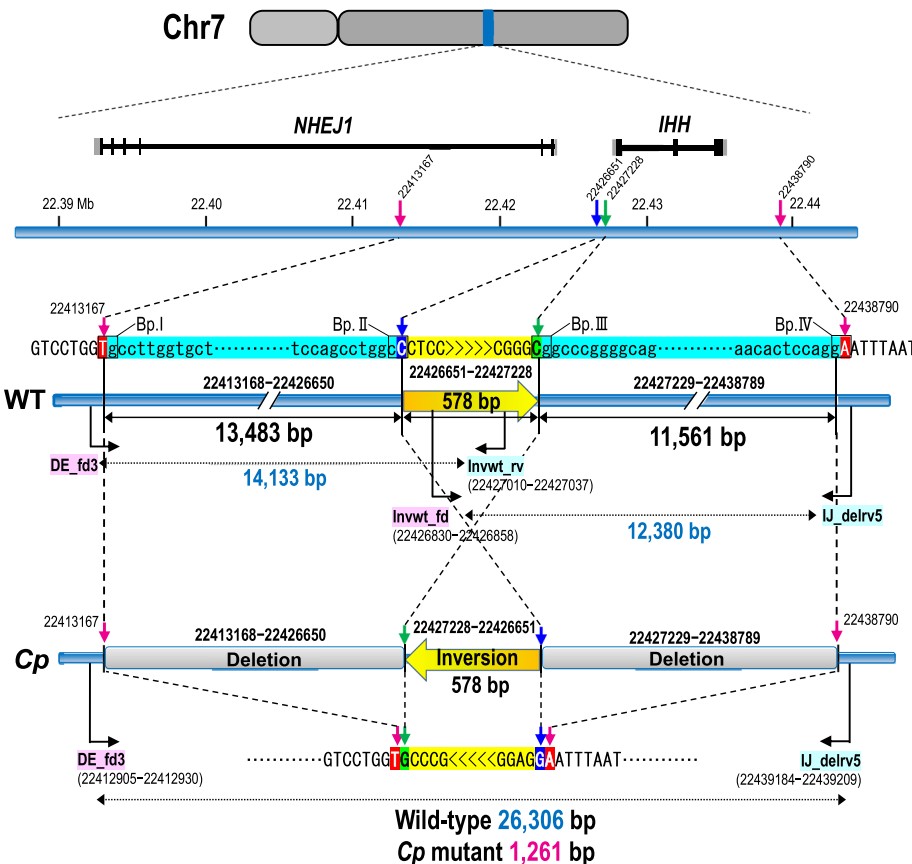

**Fig. 2 Genomic structures of the wild-type (WT) and Creeper (Cp) alleles at the Cp locus.** Two large deletions are identified at the nucleotide position 22,413,168–22,438,789: **a** 13,483 bp deletion at 22,413,168–22,426,650 and an 11,561 bp deletion at 22,427,229–22,438,789. A 578 bp fragment (yellow) is oriented in the opposite direction in the deleted region. The reference sequence of chromosome 7 was taken from the genome assembly Gallus_gallus-5.0.

Fig. 9). The 14,133 bp and 12,380 bp PCR products were only amplified in the wild-type (+/+) and Cp/+ heterozygotes, respectively, but not in the Cp/Cp homozygote (Supplementary Figs. 9 and 10b). Consequently, two structural genes, nonhomologous end-joining factor 1 (NHEJ1) and IHH, occured within the 25 kb deleted region. The breakpoints were located in the fourth intron of NHEJ1 and behind the 3′ UTR of IHH; thus, the fifth and sixth exons of NHEJ1 and the entire IHH region were deleted (Fig. 2 and Supplementary Fig. 11).

**Genotype–phenotype correlation.** The association between the Creeper phenotype and the 25 kb deletion was examined by a diagnostic PCR. The PCR primers were in the 11,561 bp deleted region (FD1: 5′-AACGATGATCTTGGTCCAAGCC-3′), the 578 bp inverted region (FD2: 5′-ACAACACCCTGATTTCAGGAGC-3′), and downstream of the deletion (RV1common: 5′-ATGCTC CCTTCCATCAATCACG-3′) (Supplementary Fig. 11a). Both the 513 bp fragment from the wild-type allele and the 396 bp fragment from the Cp allele were amplified from heterozygous (Cp/+) GSP/Cp chickens, but only the 513 bp fragment was amplified from the wild-type (+/+) GSP/Cp chickens and the 396 bp fragment from homozygous (Cp/Cp) JB embryos (Supplementary Fig. 11b). In the E2.5 – E3.0 embryos obtained from mating between Cp/+ heterozygotes of the GSP/Cp strain, one third of the live embryos showed abnormal morphogenesis (33/106) (Supplementary Table 2). A total of 82% of the abnormal embryos (27/33) were homozygous for the 25 kb deletion (del/del); most of the Cp/Cp embryos had multiple morphogenetic abnormalities, whereas abnormal wild-type (+/+) and Cp/+ embryos each had a single type of abnormality (Supplementary Table 3). The deletion was also directly associated with

the Creeper phenotype in a total of 295 F₂ progeny and 22 chicken strains and/or populations derived from 17 chicken breeds, including the red junglefowl (Supplementary Table 9). Heterozygotes expressing the 513 bp and 396 bp fragments exhibited the Creeper phenotype with short legs in the F₂ progeny and three Japanese native breeds (Japanese bantam, Miyaji-dori, Jitokko).

**Expression of IHH and NHEJ1 at early embryonic stages.** We examined the expression patterns of IHH and NHEJ1 during early embryonic development in the GSP/Cp strain. Low levels of IHH expression was detected in the neural tube and head mesenchyme in the wild-type E3 embryos and high IHH expression was detected in the autopod of the limb buds in the wild-type E6 embryos (Fig. 3a). Low levels of NHEJ1 expression was also observed in the neural tube and head mesenchyme but not in the autopod of the limb buds. In the Cp/Cp embryos, hypoplasia of the extraembryonic blood vessels was observed at E3; IHH and NHEJ1 were not expressed in the extraembryonic blood vessels and in blood islands expressing aminolevulinate dehydratase (ALAD), a marker of the blood island formation (GEISYA web site, http://geisha.arizona.edu/geisha/) (Fig. 3b, c).

**Osteoblast differentiation and chondrocyte maturation.** To understand whether the Cp/Cp cells can differentiate into osteoblasts, we isolated calvarial cells from the E15 JB strain embryos, cultured them for 16 days, and then examined the activity of alkaline phosphatase (AP). Strong AP activity was observed in the wild-type (+/+) cells, but there was little activity in the Cp/Cp cells (Fig. 4a). To confirm osteoblast differentiation in vivo, we examined the expression patterns of osteoblast differentiation

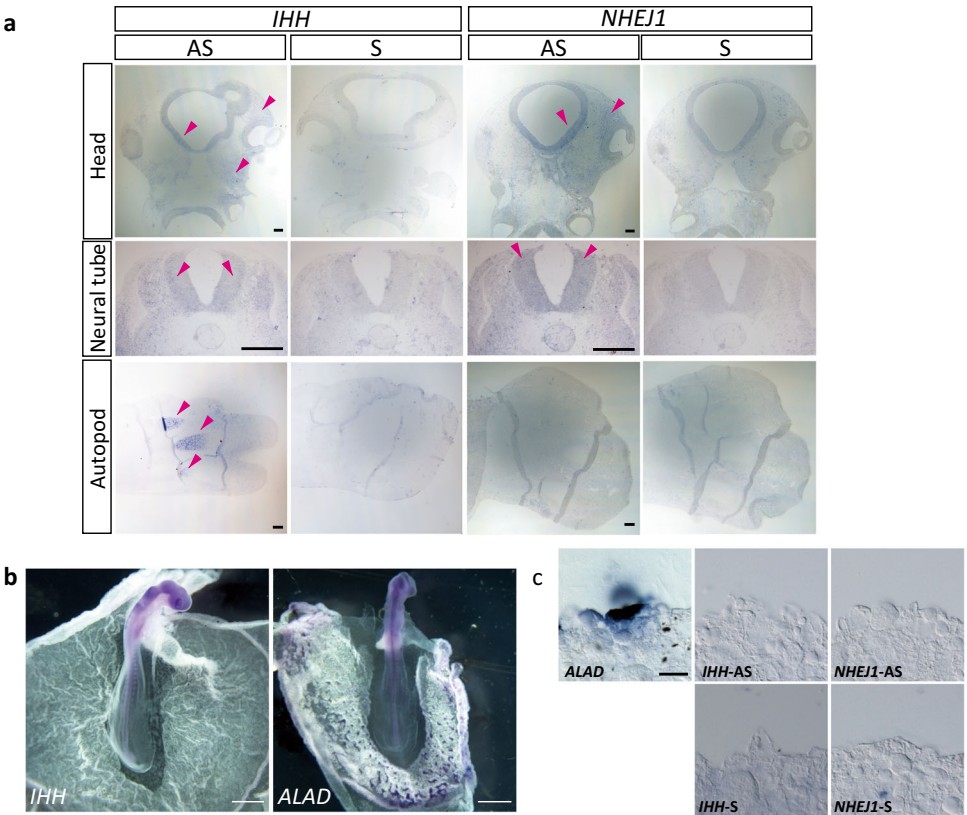

**Fig. 3 Expression of *NHEJ1* and *IHH* in embryos and extraembryonic blood vessels. a** In situ hybridization with *NHEJ1* and *IHH* in sections of the head region and neural tube of the wild-type E3 embryos and the autopod in the hindlimb of the wild-type E6 embryos. Sections are oriented with the rostral side up and the caudal side down in the head region; the dorsal side up and the ventral side down for the neural tube; and the anterior side up and the posterior side down for the autopod. Magenta arrowheads indicate gene expression. AS, antisense probe; S, sense probe. Bar, 100 μm. **b** Expression of *IHH* and *ALAD* in the wild-type embryos at stage 14, indicated by in situ hybridization. Images are oriented with the rostral side up and the caudal side down. Bar, 1 mm. **c** Neighboring sections of the extraembryonic blood vessels of embryos at stage 14. Expression of *ALAD*, *IHH*, and *NHEJ1* was examined by in situ hybridization. Bar, 200 μm.

markers in the hindlimb of the E15 embryos. Expression of a pre-osteoblast marker, runt-related transcription factor 2 (*RUNX2*), and a mature osteoblast marker, osteopontin (*OPN*), were observed in the zeugopod regions of the wild-type embryos. However, no expression of *OPN* and only faint *RUNX2* expression were found in the *Cp/Cp* embryos. Furthermore, expression of *OPN* and a hypertrophic chondrocyte marker, Type X collagen (*COL10*), were observed in the digit of the wild-type embryos, whereas only faint expression of *COL10* and *OPN* was observed in the digit of the *Cp/Cp* embryos. These results indicate that the *Cp/Cp* cells can hardly differentiate into osteoblasts due to the loss of *IHH* expression.

To elucidate the differentiation process of cartilage in the *Cp* mutant embryos, we performed in situ hybridization of *IHH* and its downstream target genes, such as patched 2 (*PTCH2*) and parathyroid hormone-related protein (*PTHrP*), and cartilage differentiation markers, type 2 collagen (*COL2*) and *COL10* on sections of the zeugopod regions in the hindlimb of the E6 embryos of the JB strain, whose *IHH* expression was down-regulated in the *Cp/+* embryos and completely deficient in the *Cp/Cp* embryos (Fig. 4b). A downstream target gene of hedgehog signaling, *PTCH2*, was still observed in the perichondrium in the *Cp/+* embryos but not in the *Cp/Cp* embryos. *PTHrP* was expressed at the end of long bones in the wild-type and *Cp/+* embryos but not in the *Cp/Cp* embryos. The cartilage was stained with Alcian blue in the *Cp/Cp* embryos, implying the possibility that cartilage cells are normally differentiated (Fig. 1) As expected, the expression of *COL2* appeared to be normal in both

the *Cp/+* and *Cp/Cp* embryos; however, *COL10* was expressed only in the wild-type embryos.

**DSB repair ability**. To elucidate the involvement of the *NHEJ1* gene in the early embryonic lethality of *Cp/Cp* homozygotes, we examined DSBs in the E3 embryos by TUNEL (terminal deoxynucleotidyl transferase dUTP nick end labeling) staining using paraffin sections (Fig. 5a). TUNEL-positive cells were found in the head mesenchyme, neural tube, and their surrounding tissues in the *Cp/Cp* embryos, but there were no TUNEL-positive cells in the heart tissue. Also, immunofluorescence staining with an anti-γH2AX (the phosphorylation on Ser139 of histone H2AX) antibody revealed strong γH2AX signaling in the mesenchymal cells of the brain and neural tube in the *Cp/Cp* embryos, indicating that DSBs accumulate concomitantly with the phosphorylation of histone H2AX (Fig. 5b). In the wild-type (+/+) and *Cp/+* embryos, neither TUNEL-positive cells nor γH2AX foci were detected.

To examine the subcellular localization of NHEJ1 in living chicken cells, we established three primary cultured cell lines from the wild-type E3 embryos, which transiently expressed an EYFP-tagged chicken NHEJ1 chimeric protein (EYFP-NHEJ1), EYFP-tagged chicken NHEJ1 mutant chimeric protein (EYFP-NHEJ1mt), or control protein (EYFP) (Fig. 6a). EYFP-NHEJ1 was present in the nuclei (Fig. 6b). EYFP-NHEJ1 accumulated at laser microirradiated sites and co-localized with a DSB marker, γH2AX, indicating that chicken NHEJ1 accumulates at DSBs (Fig. 6b, c). Time-lapse imaging showed that EYFP-NHEJ1

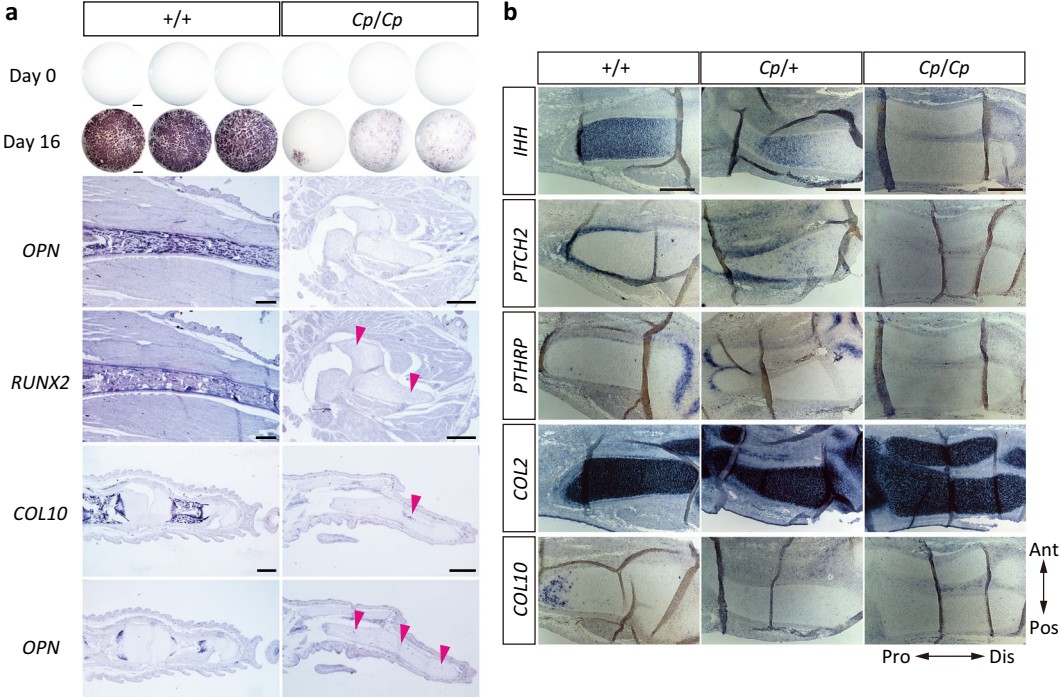

**Fig. 4 Expression of osteoblast differentiation- and chondrocyte maturation-related genes in the *Cp* mutant. a** The expression of alkaline phosphatase in calvarial cells that were isolated from the E15 wild-type (+/+) and homozygous (*Cp/Cp*) embryos and then cultured for 16 days in the differentiation medium. Bar, 1 mm. The expression patterns of *OPN* and *RUNX2* in the zeugopods of the E15 embryos and *COL10* and *OPN* in the digits. Magenta arrowheads indicate the faint expression of each gene. Bar, 500 μm. **b** The expression of *IHH*, *PTCH2*, *PTHrP*, *COL2*, and *COL10* as shown by in situ hybridization, with the neighboring sections of the zeugopod in the hindlimbs of the E6 wild-type (+/+), heterozygous (*Cp/+*), and homozygous (*Cp/Cp*) embryos from the JB strain. Ant, anterior; Pos, posterior; Dis, distal; Pro, proximal. Bar, 100 μm.

accumulated at the microirradiated sites 5 s after irradiation and the intensity of the EYFP signal increased quickly at the microirradiated sites (Fig. 6d). The mutant NHEJ1 isoform (the putative truncated protein product) located in the nucleus and cytoplasm and did not accumulate at laser-induced DSB sites (Fig. 6e). These results suggest that the mutant NHEJ1 has lost its native DSB repair ability (Fig. 6f).

To examine whether the loss of NHEJ1 function causes the embryonic lethality of the *Cp/Cp* embryos, we electroporated the full length *NHEJ1* construct into the neural tube of the E2 *Cp/Cp* embryos and then performed TUNEL staining (Fig. 5c). As a result, DSBs in the neural tube were rescued by exogenous *NHEJ1* expression, which indicates that the loss of NHEJ1 function causes the accumulation of DSBs in the head and neural tube, resulting in apoptosis and subsequent hypoplasia of the head.

**γH2AX expression after irradiation.** To investigate the DNA repair efficiency of the *Cp* mutant cells, we examined the expression of X-irradiation-induced γH2AX as a DSB marker and its elimination in the wild-type and mutant cells by confocal laser scanning microscopy using an anti-γH2AX antibody. The expression of γH2AX was normal in the *Cp* mutant because the intense γH2AX foci were detected 1 h after 2 Gy X-irradiation in cell lines from both the wild-type and *Cp/Cp* embryos (Fig. 6g). However, the elimination of γH2AX from the *Cp/Cp* cells was much slower than in the wild-type cells, although its elimination was detected 1–24 h after X-irradiation in both cell lines. The γH2AX signal persisted in the mutant cells for longer than 24 h after DSB induction, in contrast to the wild-type cells. These findings suggest that the repair efficiency of ionizing radiation (IR)-induced DSBs reduces in the mutant cells.

## Discussion

The *Cp/+* heterozygous embryos showed the shortness of fore-limbs and hindlimbs in both the JB and GSP/*Cp* strains. The *Cp/Cp* homozygous embryos of the JB strain had a higher rate of survival until the later stages of development. However, the *Cp/Cp* embryos in the GSP/*Cp* strain mostly died at the early stage of embryonic development around E3 having multiple morphogenetic abnormalities, such as hypoplasia of brain, heart, and extraembryonic blood vessels. These results suggest that genetic background, such as a partially dominant modifier(s), may influence the stage of lethality of *Cp/Cp* homozygotes[4,18,19]. Chromosome mapping and nucleotide sequence analysis of the *Cp* locus in our mutant revealed that the causative mutation is a 25 kb deletion containing the fifth and sixth exons of *NHEJ1* and the whole region of *IHH*. This differs from Chinese Xingyi bantam chickens, as no DNA fragments were amplified from the *Cp/Cp* embryos of the GSP/*Cp* strain with the PCR primers used by Jin et al.[10]; the forward primer was in the 578 bp inverted region of the 25 kb deletion in the JB and GSP/*Cp* strains. Thus, the combined *IHH* and *NHEJ1* deletions are likely involved in the early embryonic lethality of *Cp/Cp* homozygotes in the GSP/*Cp* strain.

Suppressed osteoblast differentiation was demonstrated in vitro (in cultured calvarial cells) and in vivo (in the hindlimbs and digits) by the expression analysis of osteoblast differentiation markers (alkaline phosphatase, *OPN*, and *RUNX2*). OPN is a major non-collagenous bone matrix protein (integrin-binding glycoprotein) that is highly expressed in osteoblasts, whose transcription is regulated by *RUNX2*[20]. RUNX2 is a bone-specific transcription factor that is essential for osteoblast differentiation and chondrocyte maturation; the hedgehog signal is required for *RUNX2* expression in the perichondrium of endochondral

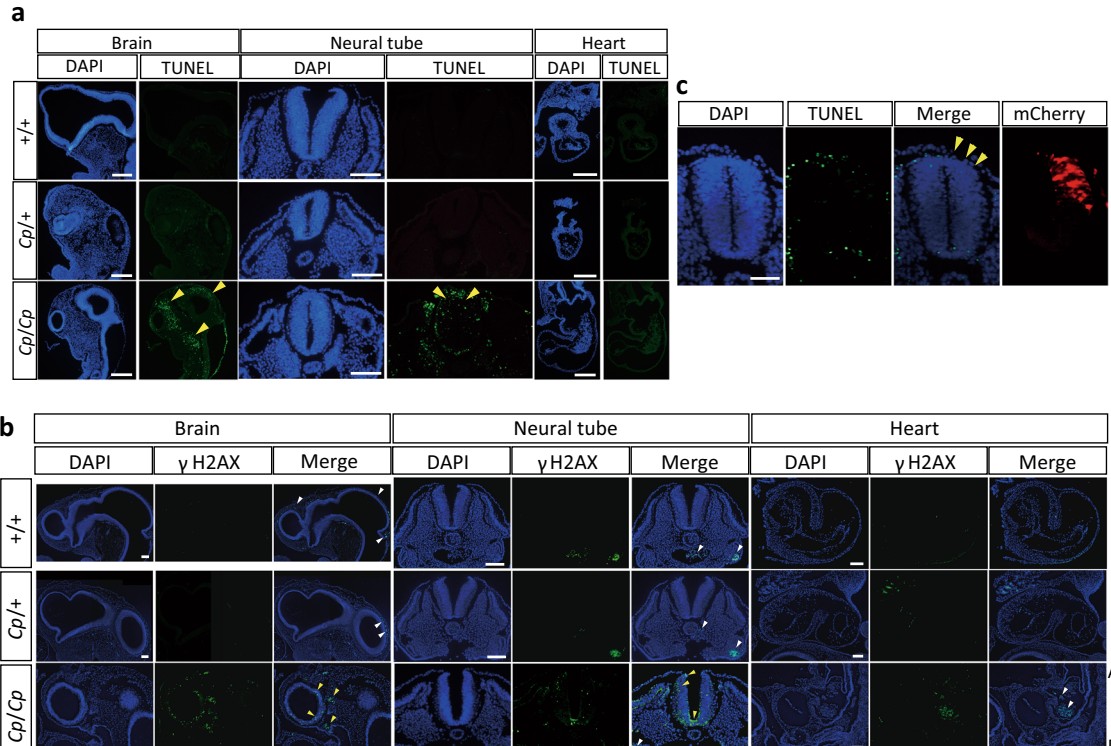

**Fig. 5 Detection of DNA double-strand breaks (DSBs) in the E3 embryos from the GSP/Cp strain. a** Detection of DSBs by TUNEL staining in the brain, neural tube, and heart. The sections are oriented with the rostral side up and the caudal side down for the brain, and the dorsal side up and the ventral side down for the neural tube. A sagittal section of the ventricle is shown for the heart. Nuclear localization was visualized with DAPI. The yellow arrowheads indicate DSBs detected by TUNEL staining. Bar, 100 μm. **b** The detection of DSBs by immunofluorescence staining with an anti-γH2AX antibody in the brain, neural tube, and heart. γH2AX signaling in the mesenchymal cells of the brain and neural tube in the *Cp/Cp* embryos are indicated with yellow arrowheads. The white arrowheads indicate non-specific fluorescence signals in the blood cells. Bar, 100 μm. **c** Exogenous *NHEJ1* expression was introduced on the right side of the neural tube by electroporation in the E2 *Cp/Cp* embryos, and DSBs were detected by TUNEL staining. The images are oriented with the dorsal side up and the ventral side down. Nuclear localization is visualized with DAPI, and the electroporated area is visualized with mCherry fluorescence. The yellow arrowheads indicate the regions where DSBs are rescued by exogenous *NHEJ1* expression. Bar, 50 μm.

bones[21], and *RUNX2* regulates the proliferation of osteoblast progenitors and their differentiation into osteoblasts via reciprocal regulations with hedgehog, *FGF*, *WNT*, and *PTHR1* signaling pathway genes[22–24]. The results presented here indicate that osteoblast differentiation is inhibited by the loss of *IHH* in the *Cp/Cp* embryos. The expression of *IHH* was downregulated in the hindlimb of the *Cp/+* embryos at E6 and completely deficient in the *Cp/Cp* embryos. *PTCH2*, a downstream target gene of SHH signaling by *IHH*[25,26] and *PTHrP*, a factor that induces osteoblast differentiation with *IHH*[13,27–29], were both expressed in the *Cp/+* embryos but not in the *Cp/Cp* embryos, indicating that the shortness of extremities is caused by the downregulation of chondrocyte proliferation. However, *COL2*, the less mature chondrocyte marker[30], was expressed in the *Cp/Cp* embryos; in contrast, *COL10*, a marker for hypertrophic chondrocytes[31,32], was not expressed in the *Cp/+* or *Cp/Cp* embryos. Chondrocyte proliferation and the subsequent differentiation to hypertrophic chondrocytes is also inhibited in the *Cp/Cp* embryos due to the loss of IHH signaling and the suppression of *PTCH2* and *PTHrP*. The same processes are delayed in the *Cp/+* embryos because of lower IHH levels, resulting in shorter limbs and smaller embryos.

DSBs are highly toxic lesions that drive genetic instability; if not repaired, the resulting chromosome discontinuity often results in cell death or cancer, and defective DSB repair is associated with developmental, immunological, and neurological disorders[33,34]. To preserve genome integrity, organisms have evolved several DSB repair mechanisms; non-homologous end-joining (NHEJ) and homologous recombination (HR) are the most prominent[35,36]. Human NHEJ1, also known as Cernunnos or XLF (XRCC4-like factor), is encoded by the *NHEJ1* gene and was discovered as an XRCC4-interacting protein by a yeast two-hybrid screening system[37,38]. The protein is mutated in patients with growth retardation, microcephaly, and immunodeficiency. NHEJ1 is a DNA repair factor essential for the non-homologous end-joining pathway, which preferentially mediates repair of DSBs. Therefore, *NHEJ1* mutation results in genetic instability, developmental delay, immunodeficiency associated with microcephaly, and increased cellular sensitivity to ionizing radiation[37,38]. In contrast, *NHEJ1* knockout mice are viable, grow normally, and are fertile[39]. Furthermore, it is noted that there is no excessive neuronal cell death or a reduction of brain weight in embryonic (E17.5) or adult (P30) *NHEJ*[−/−] mice, although the human NHEJ1 mutation is associated with microcephaly[38]. This suggests that the requirement of NHEJ1 may differ between animal species or cell types.

The *NHEJ1* knockout in the chicken DT40 B-cell line showed cellular hypersensitivity to X-rays and agents that cause DSBs[40]. However, there have been no reports of genetic diseases caused by mutations of core *NHEJ* genes, including *NHEJ1*, in birds, and the localization and function of NHEJ1 at DSB sites remain unknown for chicken cells. The present study revealed that the wild-type NHEJ1 was localized in the nuclei, immediately accumulated and formed foci at laser-induced DSBs in living chicken cells. However, the mutant NHEJ1 form of the *Cp* mutant cells has lost the

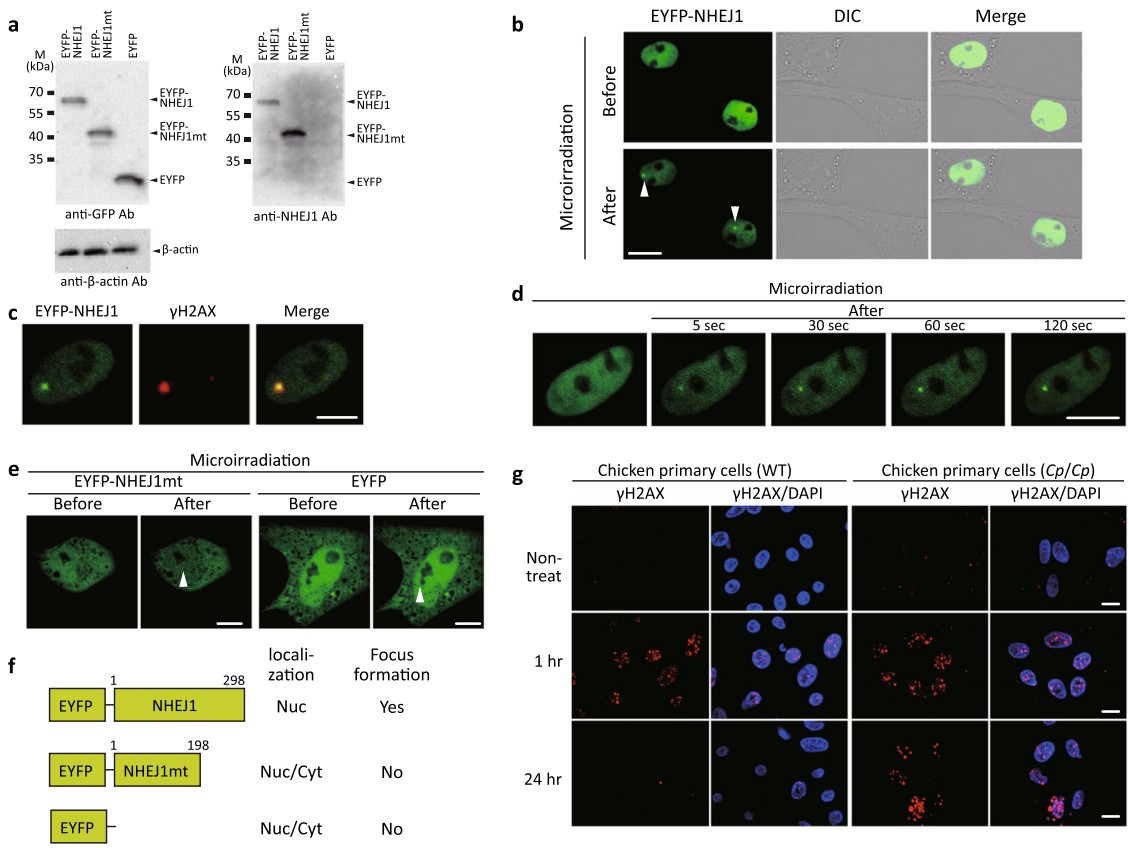

**Fig. 6 Expression of *NHEJ1* and DSBs in the *Cp* mutant. a** Expression of an EYFP-tagged chicken *NHEJ1* gene and an EYFP-tagged *Cp*-derived mutant *NHEJ1* gene in chicken primary cells. Extracts from the wild-type chicken primary cells transiently expressing the EYFP-chicken NHEJ1 chimeric protein (EYFP-NHEJ1), EYFP-chicken NHEJ1 mutant chimeric protein (EYFP-NHEJ1mt), or control protein (EYFP) were prepared and subjected to Western blotting using anti-GFP, anti-NHEJ1, or anti-β-actin antibodies. **b** Imaging of the EYFP-NHEJ1-transfected chicken primary cells before and after 405 nm laser microirradiation. Expression of EYFP-NHEJ1, differential interference contrast images (DIC), and merged images are shown. EYFP-NHEJ1 accumulates immediately at DSBs induced by laser microirradiation. Arrowheads show the microirradiated sites. **c** Immunostaining of microirradiated EYFP-NHEJ1-expressing cells with an anti-γH2AX antibody. The cells were fixed and stained with the anti-γH2AX antibody 5 min after microirradiation. Expression of EYFP-NHEJ1, immunostaining with the anti-γH2AX antibody, and merged image are shown. **d** Time-dependent EYFP-NHEJ1 accumulation in a living cell 5–120 s after microirradiation. **e** Primary cells that express EYFP-NHEJ1mt or EYFP before and after microirradiation. Arrowheads show the microirradiated sites. The EYFP-NHEJ1mt is mislocalized in cells and cannot accumulate at DSB sites. **f** Summary of the results presented. Nuc, nucleus; Cyt, cytoplasm. **g** Comparison of γH2AX expression in the wild-type (+/+, WT) and mutant (*Cp/Cp*) chicken cells from the GSP/*Cp* line. The cells were X-irradiated with 2 Gy, and immunostaining patterns of γH2AX were examined 1 h and 24 h after X-irradiation. Nuclear DNA was stained with DAPI. The γH2AX expression remains even after 24 h in the mutant (*Cp/Cp*) chicken cells. All scale bars indicate 10 μm.

normal localization and DNA-binding abilities that are necessary to repair DSB; therefore, DSB repair activity, i.e., NHEJ activity, is impaired in the *Cp* mutant cells. We previously reported that the amino acid sequences in the C-terminal region (CTR) of NHEJ1, containing the nuclear localization signal, are highly conserved among domestic animals, including the chicken[41]. This suggests that the CTR is essential for NHEJ1 functioning[41]. The CTR of NHEJ1 contains the nuclear localization signal and the Ku binding motif, and the recruitment and retention of NHEJ1 at DSBs in cells requires Ku interaction[41–44]. Our results show that the truncated NHEJ1 is not recruited to damaged sites but that it is partially localized in the nucleus. Together, our findings support the importance of the Ku binding motif in the CTR of NHEJ1 for the recruitment and retention of XLF at DSBs. The γH2AX signal persisted in the *Cp/Cp* cells longer than 24 h after DSB induction by irradiation, indicating that the repair efficiency of IR-induced DSBs is reduced in the mutant cells. *NHEJ1* expression was observed in the neural tube and head mesenchyme of the wild-type E3 embryos from the GSP/*Cp* strain, similar to the human adult central nervous system (CNS)[45], but not in the autopod of limb buds. However, *IHH* expression was

observed in the autopod of limb buds, suggesting that osteosclerosis in the *Cp* mutant is mainly caused by the loss of *IHH*. We found typical TUNEL-positive cells, which result from the accumulation of unrepaired DSBs, in the head mesenchyme, neural tube, and its surrounding tissues in the *Cp/Cp* embryos, but not in the heart tissue. DSBs in the neural tube of the *Cp/Cp* embryos were rescued by electroporation of a full length exogenous *NHEJ1*. These results suggest that the head region and neural tube of chick embryos at the early stage are sensitive to DSBs and that the loss of *NHEJ1* function causes the accumulation of DSBs in these tissues of the *Cp/Cp* embryos, leading to apoptosis.

Hypoplasia of extraembryonic blood vessels was observed in the *Cp/Cp* embryos at E3[46]. This phenotype appears to be reminiscent of the circulatory abnormalities reported in *Ihh* knockout mice[12]. Half of the *Ihh*$^{-/-}$ mouse embryos die between E10.5 and E12.5, which is most likely due to circulatory abnormalities caused by the lack of IHH activity in the yolk sac, whereas the other half can survive to birth. However, hypoplasia of the brain or heart was observed before circulation starts in our *Cp/Cp* embryos, indicating that these abnormalities cannot be

explained by the loss of IHH activity alone. *IHH* and *NHEJ1* were not expressed in the extraembryonic blood vessels where *ALAD* was expressed in chick embryos; therefore, the hypoplasia of extraembryonic blood vessels in the *Cp/Cp* embryos could be due to the secondary effect of the loss of *IHH* and/or *NHEJ1*, which accelerates mortality during the early stages of embryonic development. The *Nhej1*[−/−] mice are not embryonically lethal[47] and show no excessive neural cell death[38,39,48]; however, *Nhej1/Paxx* double knockout mice have severe neuronal apoptotic events in the brain that lead to embryonic mortality, which is associated with genomic instability[48,49]. In contrast, human and chickens present neural cell death and/or neuronal organization disorders because of a mutation of the *NHEJ1* gene alone[38,45]. Thus, the repair deficiency of DSBs that occurs in the brain and neural tube is the main cause of the early embryonic lethality in our *Cp* mutant, and *NHEJ1* plays a pivotal role in DSB repair in chick embryos, similar to that in humans. This is also supported by the result that the apoptosis of the neural tube was rescued by the exogenous expression of a full length *NHEJ1* gene in the *Cp/Cp* embryos.

This study demonstrates that the loss of IHH signals causes abnormal chondrocyte proliferation and differentiation and inhibits osteoblast differentiation in the Creeper chicken. Embryonic lethality in the *Cp/Cp* embryos can be explained by impaired DSB repair, concomitant with the suppression of cell growth and abnormalities in the vascular and nervous systems due to the functional loss of *IHH* and *NHEJ1*. Our new *Cp* mutant could become a good animal model for human *NHEJ1* disease to examine the function of non-homologous end-joining factors and the molecular basis of blood vessel formation involving *IHH* for normal embryonic development.

## Methods

**Ethics statements**. Animal care and all experimental procedures were approved by the Animal Experiment Committee, Graduate School of Bioagricultural Sciences, Nagoya University (approval no. 2014021406). Experiments were conducted according to the Regulations on Animal Experiments at Nagoya University.

**Animals and mating**. The following three chicken strains were used for this study: the JB strain derived from the Japanese bantam breed "Chabo" that has the Creeper (*Cp*) trait (Supplementary Fig. 1); and the GSP and PNP/DO strains, both of which are highly inbred strains derived from the Fayoumi breed[17,50,51]. All lines are maintained at the Avian Bioscience Research Center (ABRC), Nagoya University, Japan. To elucidate the molecular mechanism that causes the *CP* phenotype, we established a congenic strain (GSP/*Cp*) of the *Cp* gene by backcrossing the *Cp*/+ heterozygous F₁ male progeny, which was obtained from mating JB males with GSP females, to GSP females and then conducting backcross mating with the GSP strain eight times (Supplementary Fig. 2). Then this congenic strain has been maintained as a closed colony. Segregation of the Creeper (*Cp*/+) and wild-type (+/+) chickens was examined in the F₁, F₂, and/or backcross generations, which were obtained from mating the JB strain with two wild-type strains (PNP/DO and GSP) (Supplementary Table 1).

**Skeletal staining of embryos**. The embryos at E14 were dissected in phosphate-buffered saline (PBS) and fixed in 70% EtOH overnight. The embryos were stained with 0.03% Alizarin red/0.01% Alcian blue in a 10% acetic acid solution overnight[52]. After washing with 1% KOH for 24 h, they were treated in 10%, 20%, 30%, 50%, and 80% glycerol made in 1% KOH diluted with distilled water, and then in 100% glycerol.

**Chromosomal localization and detection of mutations**. To determine the chromosomal location of the *Cp* locus, SNP genotyping was conducted with pooled DNA samples from wild-type homozygotes (+/+) in the GSP strain (*n* = 16), and pooled samples from heterozygotes (*Cp*/+) in the GSP/*Cp* strain at the N₆ generation (*n* = 19). Blood samples were collected from the wing vein of adult or young GSP chickens and heart tissues were collected from Creeper chicks 1 day after hatching. Genotyping was performed using a 600 K Axiom® Chicken Genotyping Array containing 580,961 SNPs (Affymetrix, Santa Clara, CA, USA). Genetic linkage mapping of the *Cp* locus was performed using 175 F₂ progeny produced from mating between four female and two male F₁ progeny that were obtained from a cross of one GSP female (+/+) and one heterozygous (*Cp*/+) JB

male (Supplementary Table 1). Blood samples were obtained from the wing vein of adult chickens in parental and F₁ generations and hearts of 1-day-old F₂ progeny.

To detect the mutation at the *Cp* locus in the JB strain, the genomic structures at the *Cp* locus were compared among the following samples: (1) a wild-type (+/+) GSP adult female; (2) a heterozygous (*Cp*/+) JB adult male; (3) a 14-day homozygous (*Cp*/*Cp*) JB embryo; and (4) a 1-day-old heterozygous (*Cp*/+) N₂ progeny produced from mating between GSP females and F₁ males that were obtained from mating a GSP female with a JB male. Genomic DNA was obtained from the blood of the wild-type (+/+) and heterozygous (*Cp*/+) chickens and liver tissues from homozygous (*Cp*/*Cp*) embryos using a phenol/chloroform extraction procedure.

**Verification of osteoblast differentiation**. Calvarial cells were isolated from the wild-type (+/+), heterozygous (*Cp*/+), and homozygous (*Cp*/*Cp*) E15 embryos of the JB strain as described by Li et al.[53]. Minced skull pieces were treated with an enzyme mixture containing 1.5 unit/ml collagenase P (Sigma-Aldrich, St. Louis, MO, USA) and 0.05% trypsin for 15 min by four sequential digestion. Two to four fractions of the cells were pooled, and an equal volume of 100% FBS (fetal bovine serum) was added to terminate the enzyme activity. The cells were cultured in DMEM (Dulbecco's Modified Eagle Medium) containing 10% FBS by plating 4 × 10⁴ cells/well in a 96-well culture plate. After the cells reached confluence, the medium was changed to a differentiation medium, αMEM (Minimum Essential Medium Eagle–Alpha Modification) containing 10% FBS, 50 mg/ml ascorbic acid, and 4 mM β-glycerophosphate. The cells were cultured for 16 days, and the medium was changed every other day until the cells differentiated into osteoblasts. Alkaline phosphatase activity was detected in a NTMT (NaCl, Tris-Cl, MgCl₂, Tween-20) solution containing 100 mM Tris pH 9.5, 100 mM NaCl, 50 mM MgCl₂, 0.1% Tween-20, 4.5 μl/ml of nitro blue tetrazolium chloride (NBT; Sigma-Aldrich), and 3.5 μl/ml of 5-Bromo-4-chloro-3-indolyl phosphate (BCIP; Sigma-Aldrich).

**In situ hybridization on embryos**. The following probes were used for in situ hybridization: *IHH* (accession no. LC278390), *NHEJ1* (accession no. LC278391) for the E3 and E6 embryos, *RUNX2*, *COL10*, *PTCH2*, *PTHrP*, and *COL2* for the E6 embryos, and *OPN*, *RUNX2*, and *COL10* for the E15 embryos. The embryos were fixed in 4% paraformaldehyde overnight and dehydrated with methanol. The embryos were then hydrated with a graded series of 75%, 50%, and 25% methanol/PBT (0.1% Tween-20 in PBS) for 5 min each and washed with PBT twice at room temperature (RT). The embryos were treated with a 6% hydrogen peroxide/PBT solution for 15 min, followed by a 10 μg/ml proteinase K solution for 5 min at RT, and fixed with 0.2% glutaraldehyde/4% paraformaldehyde for 20 min. The embryos were hybridized with specific probes in a hybridization buffer (25 ml formamide, 12.5 ml of 20x SSC pH 5.0, 625 μl of 50 mg/ml tRNA, 100 μl of 500 mg/ml heparin, 10 ml 10% SDS, 2.5 ml H₂O) at 70 °C overnight. The embryos were washed with a solution of 25 ml formamide, 5 ml of 20x SSC pH 5.0, 5 ml of 10% SDS, and 15 ml H₂O and then treated in blocking solution containing 20% horse serum in 0.1% Tween/TBS (Tris-buffered saline). After anti-Digoxigenin-AP antibody treatment overnight at 4 °C, the embryos were washed four times with a 0.1% Tween/TBS solution for 30 min. Finally, the embryos were incubated in a detection solution (1 ml of 5 M NaCl, 5 ml of 1 M Tris-HCl pH 9.5, 2.5 ml of 1 M MgCl₂, 2.5 ml of 20% Tween-20, 39 ml H₂O, and 175 μl of 100 mg/ml NBT and 175 μl of 50 mg/ml BCIP)[54].

**TUNEL staining**. TUNEL staining was performed using the In Situ Cell Death Detection Kit, Fluorescein (Roche Diagnostics, Basel, Switzerland) following the manufacturer's instructions. The E3 embryos were harvested in ice-cold PBS and fixed in 4% paraformaldehyde/PBS overnight. Embryos were embedded in paraffin and sectioned with a Leica microtome to produce 7 μm sections. Sections were put through a hydration process and treated with 10 μg/ml proteinase K solution for 15 min at 37 °C to promote permeabilization. The TUNEL reaction was performed at 37 °C for 1 h in the humidified chamber. Then, 1 μg/ml 4, 6-diamino-2-phenylindole (DAPI) solution was added to each section and incubated for 10 min before being mounted in DAKO fluorescent mounting medium (Agilent Technologies, Santa Clara, CA, USA).

**Cell culture and gene transfection**. Fibroblast cells were prepared from the E3 embryos from the wild-type (+/+), heterozygous (*Cp*/+), and homozygous (*Cp*/*Cp*) individuals from the GSP/*Cp* strain. The cells were cultured in 199 medium (Thermo Fisher Scientific-Gibco, Carlsbad, CA, USA) supplemented with 18% FBS and 60 μg/ml kanamycin sulfate (Sigma-Aldrich) at 39 °C in 5% CO₂ in the air.

The complementary DNA (cDNA) fragments were synthesized from the wild-type chicken *NHEJ1* gene (298 amino acids) (accession no. LC339852) and the *NHEJ1* mutant gene derived from the *Cp* allele (198 amino acids) (accession no. LC339853), which have an artificial *Eco*RI site at the 5′ end and *Bam*HI site at the 3′ end. The fragments were ligated at the *Eco*RI and *Bam*HI sites in the pEYFP-C1 vector (Clonetech, Palo Alto, CA, USA) to produce in-frame fusion genes. The pEYFP-chicken *NHEJ1*, pEYFP-chicken *NHEJ1* mutant, or pEYFP-C1 was transiently transfected into cells using Lipofectamine 3000 (Invitrogen, Carlsbad, CA, USA) according to the manufacturer's instructions. The cells were cultured for

2 days and then monitored under an FV300 confocal laser scanning microscope (CLSM) (Olympus, Tokyo, Japan).

**Immunoblotting**. Total cell lysates were electrophoresed on 5–20% SDS-polyacrylamide gels, and the fractionated products were electroblotted onto Hybond-P membranes (GE Healthcare Bio-Sci. Corp., Piscataway, NJ). The membranes were blocked in Blocking One (Nacalai Tesque, Kyoto, Japan) or ECL Prime Blocking reagent (GE Healthcare Bio-Sciences, Piscataway, NJ, USA) for 30 min at RT[55,56]. The following antibodies were then applied: mouse anti-XLF monoclonal antibody (D-1, 1:500) (Santa Cruz Biotechnology, Santa Cruz, CA, USA), rabbit anti-GFP polyclonal antibody (FL, 1:4000) (Santa Cruz Biotechnology), or mouse β-actin monoclonal antibody (AC-15, 1:500) (Sigma-Aldrich). The antibodies were diluted in Signal Enhancer HIKARI (Nacalai Tesque). Binding to each protein was detected using the Select Western blotting detection system (GE Healthcare Bio-Sciences) and visualized using the ChemiDoc XRS system (Bio-Rad, Hercules, CA, USA).

**Immunofluorescence staining**. Immunofluorescence staining for γH2AX detection was performed for tissue sections of the E3 embryos and the above-mentioned cultured fibroblast cells[56,57]. The sections and fixed cells were blocked using a blocking solution and then incubated for 30 min at RT with a mouse anti-γH2AX monoclonal antibody (JBW301) (Upstate Biotechnology, Charlottesville, VA, USA). After washing with PBS, detection was performed using an Alexa Fluor 468- or 568-conjugated secondary antibody (Molecular Probes, Eugene, OR, USA). DNA was stained with 0.025 μg/ml DAPI solution (Sigma-Aldrich).

**Electroporation in the neural tube of embryos**. A CMV-derived human *NHEJ1* construct[56] was diluted in distilled water at a concentration of 2 μg/μl. The DNA solution was injected into the spinal cord of the embryos at stage 15 using a glass pipet. The pCAGGS-mCherry (Takara Bio, Kusatsu, Japan) (0.8 μg/μl) was co-electroporated into the site to visualize the electroporated area. Five electric pulses at 17 V lasting 25 ms were delivered to the right side of the neural tube by a pair of platinum electrodes with a CUY21-EDIT electroporator (BEX, Tokyo, Japan).

**X-irradiation, local DNA damage induction, and cell imaging**. The cells were exposed to 2 Gy of X-rays at a dose rate of 0.8 Gy/min at RT. The X-rays were generated at 200 kVp/20 mA and passed through 0.5 mm Cu and Al filters using Pantak HF320S (Shimadzu, Kyoto, Japan)[57]. The local DSBs were induced using a 3–10% power scan with a 405 nm diode laser (Olympus, Tokyo, Japan) for 1 s[55,56]. Images of living or fixed cells expressing EYFP-chicken *NHEJ1*, the EYFP-chicken *NHEJ1* mutant, or EYFP alone were obtained using a confocal scanning laser microscopy system (FV300 CLSM, Olympus).

**Statistical analyses and reproducibility**. Each individual experiment was repeated at least three times, and similar results were obtained. The chi-square test was used to test that the *Cp* allele segregates on Mendelian principles in F₁, F₂, and backcross progeny. The shank lengths of the wild-type and Creeper chicks were compared with the Welch's *t*-test was used to compare. A $P < 0.05$ was considered as significant.

**Reporting summary**. Further information on research design is available in the Nature Research Reporting Summary linked to this article.

## Data availability

The DNA sequence data have been deposited to the DNA Data Bank of Japan (DDBJ); the 1031 bp sequence that includes the 578 bp inverted sequence in the deleted region (accession no. LC332495); and the sequences of cDNA fragments of the wild-type and CP mutant *NHEJ1* genes (accession nos. LC339852 and LC339853, respectively). All other data are present in the manuscript and its supplementary files.

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

## Acknowledgements

We thank Dr. Akira Ishikawa, Nagoya University for linkage mapping of the *Cp* locus, Dr. Masaoki Tsudzuki, Hiroshima University, Japan for providing DNA samples of Japanese indigenous chicken breeds, Mr. Yasutomo Yutoku, National Institute of Radiological Sciences for technical assistance, and Dr. John F. Fallon, University of Wisconsin-Madison, USA for providing cDNA fragments of the *PTCH2, PTHrP, COL2, and COL10* genes. The JB and GSP/*Cp* strains were provided by the Avian Bioscience Research Center, Graduate School of Bioagricultural Sciences, Nagoya University through the National Bio-Resource Project (NBRP) "Chicken/Quail" supported by the Ministry of Education, Culture, Sports, Science, and Technology (MEXT) and Japan Agency for Medical Research and Development (AMED), Japan. This work was supported by Grant-in-Aid for Scientific Research on Innovative Areas (23113004), MEXT to Y.M., and Grant-in-Aid for Scientific Research (B) (18H02487), Inamori Foundation, Mochida Memorial Foundation for Medical and Pharmaceutical Research, and The NOVARTIS Foundation (Japan) for the Promotion of Science, Japan, to T.S.

## Author contributions

K.K., T.S., M.K., and Y.M. conceived this study and designed the experiments. K.K. and T.U. performed genome mapping and identification of causative mutations. T.S. performed developmental and histological observation of embryos, in situ hybridization, and gene transfer into embryos. M.K. and A.K. performed gene transfer into cultured cells, immunoblotting, and immunocytochemical analyses of irradiated cells. C.N. established cell lines. K.K., M.N., and K.I. performed gene mapping with SNP arrays. K.K., T.S., M.K., and Y.M. wrote the manuscript with comments from the other authors. All authors have reviewed the manuscript.

## Competing interests

The authors declare no competing interests.
