## [Peer Review File · Communications Biology]

Reviewers' comments:

Reviewer #1 (Remarks to the Author):

This fascinating study by Kinoshita et al describes elegant characterisation and functional validation of a genetic deletion involving both the *Ihh* and *NHEJ1* genes in an unconventional model, the creeper chicken. The data generated are generally novel, of high standard and will be of interest to a broad audience ranging including those studying chondrodystrophy, developmental biologists, and those studying genomic stability. However a few major concerns need to be addressed:

1) The second result section claims "Osteoblast differentiation is inhibited in the Cp mutant." The data provided in this section relates to abnormal shape and size of various long bones and the lack of mineralised tissue formation. None of this directly assesses osteoblast differentiation, for example by detecting transcripts of osteoblast differentiation markers or differentiating osteoblasts to form mineralised nodules in vitro. Failure of osteoblast proliferation or activity could equally explain these phenotypes. Osteoblast differentiation should be directly assessed.

2) The conclusion in the same section that "chondrocyte proliferation and subsequent differentiation are completely inhibited" (repeated in the discussion) is similarly not supported by direct evidence, for example histological analysis of chondrocyte proliferation. The authors subsequently show absence of the hypertrophic chondrocyte marker COL10, so an amendment of this conclusion may be sufficient.

3) Localisation of *IHH* and *NHEJ1* in normal embryos shown in Figure 3 is exceedingly weak. The *NHEJ1* sense and anti-sense probes appear to have produced comparable staining in the neural tube in the images provided. These localisations need to be improved or replaced with protein-level analyses.

4) Interpretation of nick-end labelling (TUNEL assay) indicating double strand breaks is confusing as this assay is primarily used to detect apoptotic cells. It is very surprising that no apoptotic cells were observed in the +/+ embryos given this process is common during development (e.g. over the closed neural tube). γ H2AX staining of cp/cp versus +/+ embryonic tissue is needed to confirm that the increase in TUNEL staining is not simply due to an increase in apoptosis in dyeing embryos.

5) Quantification and statistical comparison of imaging data (e.g. % TUNEL positive cells within rescue and control regions in Figure 4C, γ H2AX staining in Figure 5g, etc) is expected throughout to demonstrate reproducibility and variability.

Minor comments:

- Line 97: "NHFJ1" should read *NHEJ1*.

- Please make sure all abbreviations and gene names are full explained (e.g. GSP, ALAD, etc).

- Lines 219-223 of the discussion seem to assume chondrocyte to osteoblast trans-differentiation as the main mode of long bone ossification ("differentiation pathway from chondrocytes to osteoblasts"). Without having directly assayed this as well as the canonical mode of endochondral ossification through blood vessel ingression, this discussion should be removed.

- A schematic comparison of the known *NHEJ* pathway similarities/differences between humans, mice and chickens would be helpful.

- Lines 285-286 claim the Cp model "highlights the importance of *NHEJ* function I normal neurogenesis". The production of neurones has not been assessed (even if reduced, apoptosis of the neuroepithelium before the onset of neurogenesis would be likely to limit neuron production). This conclusion should be removed.

Reviewer #2 (Remarks to the Author):

This is a lovely manuscript, well written and clearly presented that deserves to be published.

Below are the points to be addressed:

Major

1) From this work, the conclusion that NHEJ1 deficiency "alone" causes embryonic lethality in the Creeper chicken can be suggested but not directly made. Strict demonstration would require a chicken with only the NHEJ1 defect.

Therefore the title of the manuscript should be "Combined deletions of IHH and NHEJ1..."

Minor

1) P 5 line 95. NHEJ1.....NHEJ1

2) Would it be possible to perform clonogenic survival assays after DSB induction (IR or radiomimetic drugs) with the established primary cells to further support the NHEJ defect.

3) P 10 line 226 discovery of NHEJ1....it is a co-discovery by ref 27 and 28

4) P 10 Lines 236-237. The CTR of NHEJ1 contains the NLS but also the Ku binding motif. Recruitment and retention of XLF at DSBs in cells requires Ku interaction see and refer to the work of Charbonnier, Caldecott and Chen. The data in this manuscript clearly show that truncated XLF is not recruited to sites of damage but that it is still can be localized in the nucleus.

5) P 12 line 281. Remove "was"

Reviewer #3 (Remarks to the Author):

In this manuscript Kinoshita et al explore the genetic and developmental origin behind the Creeper chicken phenotype. They discover that the phenotype is linked to a double deletion/inversion impairing the functions of two genes: IHH and NHEJ1. While the implication of IHH in the phenotype is well described, the implication of NEJ1 is rather new. Using expression analysis and functional experiments they gather arguments indicating that the loss of function of NEJ1 is causing the early mortality phenotype of homozygous creeper, a part of the phenotype that is not easily explained by the loss of IHH.

This manuscript addresses a classical genetic problem: identifying genes causing a well-characterized phenotype using chicken as a model system. Because the work complements and rectifies nicely what has been published on the problem (i.e. the implication of IHH in the Creeper phenotype), this manuscript is of importance for the field and should be published in Communications Biology. However several experiments are lacking and there are some flaws in the manuscript. I will list them hoping that the authors will be able to address them to improve their work and be able to publish it:

Main concerns:

1. The morphogenetic abnormalities of the early stages of Creeper embryos (title of the first part of the results) are not well characterized. That is an important piece of result that should be in the main figures. The only information are in Suppl Fig 3G where we can see a Tunnel staining in a mutant but not control embryos and a descriptive table: hypoplasia of heart and brain and abnormal island formation should be better documented (mutants versus control embryos).

2. The description of the deletions are very informative and important to this manuscript, however to be sure that the same deletions are causing the initial Creeper phenotype it will be informative to double check that these deletions exist in the JB and/or the Chinese strains too.

Minor concerns:

1. There is no clear rationale (in the main) text for choosing the GSP strains to study the mutation.
2. There is no explanation on why ALAD gene is being investigated and not another gene.
3. A bit of introduction on DSB would be welcome for the readers.
4. The panels of the figures are very small and are difficult to see (in particular Fig. 3 and Fig. 4).
5. Sentence starting at line 205 (discussion) is confusing: the deletion is likely the same in the Chinese strain (see main concern 2). Are they potentially different genetic mutation causing the Creeper phenotype? This is unclear to me.

Responses to the comments of the reviewers

Reviewer #1 (Remarks to the Author):

1) The second result section claims “Osteoblast differentiation is inhibited in the Cp mutant.” The data provided in this section relates to abnormal shape and size of various long bones and the lack of mineralised tissue formation. None of this directly assesses osteoblast differentiation, for example by detecting transcripts of osteoblast differentiation markers or differentiating osteoblasts to form mineralised nodules in vitro. Failure of osteoblast proliferation or activity could equally explain these phenotypes. Osteoblast differentiation should be directly assessed.

Line 144 – 158, Fig. 4a: In order to investigate osteoblast differentiation in the *Cp* mutant cells *in vitro*, we isolated calvarial cells of E15 embryos from the JB strain, cultured them for 16 days and then examined alkaline phosphatase activity, an osteoblast differentiation marker. The alkaline phosphatase activity was found in the wild-type (+/+) cells, whereas the activity was very weak in the *Cp/Cp* cells. Next, to confirm osteoblasts differentiation *in vivo*, we examined gene expression of a pre-osteoblast marker, *RUNX2*, and a mature osteoblast marker, *osteopontin (OPN)* in the hindlimb of the E15 embryos. *RUNX2* and *OPN* were strongly expressed in the zeugopod region of the wild-type embryos; however, faint and no expression were found for *RUNX2* and *OPN*, respectively, in the *Cp/Cp* embryos. Also in the digits, *OPN* expressed strongly in the wild-type but faint in the *Cp/Cp* embryos. These results clearly indicate that osteoblast differentiation is inhibited in the *Cp/Cp* embryos due to the loss of *IHH* expression.

2) The conclusion in the same section that “chondrocyte proliferation and subsequent differentiation are completely inhibited” (repeated in the discussion) is similarly not supported by direct evidence, for example histological analysis of chondrocyte proliferation. The authors subsequently show absence of the hypertrophic chondrocyte marker COL10, so an amendment of this conclusion may be sufficient.

Line 159 – 175 and Fig. 4b: We have demonstrated that chondrocyte proliferation and subsequent differentiation is also inhibited in *Cp/Cp* homozygotes due to the loss of *IHH* expression, resulting in shorter limbs and smaller sized embryos, by expression analysis of its downstream target genes of *IHH*, *PTCH2* and *PTHrP*, and cartilage differentiation markers, *COL2* and *COL10*, in the zeugopod regions of hindlimbs of the E6 *Cp/+* and *Cp/Cp* embryos. We also checked expression of *COL10* and *osteopontin* at later stage in the E15 *Cp/Cp* embryos, and their expressions were strongly downregulated. These results suggest that the chondrocyte proliferation and subsequent differentiation to hypertrophic chondrocytes is also inhibited in

the *Cp/Cp* embryos because of the loss of IHH signaling as well as the inhibition of osteoblast differentiation.

3) Localisation of IHH and NHEJ1 in normal embryos shown in Figure 3 is exceedingly weak. The NHEJ1 sense and anti-sense probes appear to have produced comparable staining in the neural tube in the images provided. These localisations need to be improved or replaced with protein-level analyses.

We could not obtain IHH and NHEJ1 antibodies that react with chicken cells; therefore, we could not accomplish this experiment. However, we re-examined *in situ* hybridization of *IHH* and *NHEJ1* with antisense and sense DIG probe, and we got the same result in which the sections stained with antisense probe had higher intensity compared to that stained with sense probe. Fig. 3a has been magnified to make it easy to see the signals.

4) Interpretation of nick-end labelling (TUNEL assay) indicating double strand breaks is confusing as this assay is primarily used to detect apoptotic cells. It is very surprising that no apoptotic cells were observed in the +/+ embryos given this process is common during development (e.g. over the closed neural tube). γ H2AX staining of cp/cp versus +/+ embryonic tissue is needed to confirm that the increase in TUNEL staining is not simply due to an increase in apoptosis in dying embryos.

Line 182 – 187 and Fig. 5b: We have also performed immunofluorescence staining with anti- γ H2AX antibody on the histological sections of the wild-type (+/+), *Cp/+* and *Cp/Cp* embryos at E3. The γ H2AX signals were observed in TUNEL-positive apoptotic cells in the neural tube and brain in the *Cp/Cp* embryos but not in the heart, which demonstrates that the accumulation of unrepaired DSBs is the main cause of apoptosis.

5) Quantification and statistical comparison of imaging data (e.g. % TUNEL positive cells within rescue and control regions in Figure 4C, γ H2AX staining in Figure 5g, etc) is expected throughout to demonstrate reproducibility and variability.

Fig. 5c and Fig. 6g: The difference of the number of γ H2AX foci was very distinct in fibroblast cells 24 hr later after irradiation between the wild-type (+/+) and *Cp/Cp* embryos. The TUNEL-positive cells were hardly observed in the part of the *Cp/Cp* embryos where the NHEJ1 cDNA was electroplated, compared with the control with no treatment. Thus we don't think quantification comparison by imaging is necessary for these data. The same experiments were repeated for the electroporation of NHEJ1 cDNA to the E6 embryos three times and for the

irradiation to the embryonic fibroblast cells twice, and the same results were obtained for all the experiments.

Minor comments:

- Line 97: “NHFJ1” should read NHEJ1.

Line 101: “NHFJ1” has been corrected to “NHEJ1”.

- Please make sure all abbreviations and gene names are full explained (e.g. GSP, ALAD, etc).

Line 76 and 140: Full names of all the abbreviations including GSP and ALAD have been described.

- Lines 219-223 of the discussion seem to assume chondrocyte to osteoblast trans-differentiation as the main mode of long bone ossification (“differentiation pathway from chondrocytes to osteoblasts”). Without having directly assayed this as well as the canonical mode of endochondral ossification through blood vessel ingression, this discussion should be removed.

Line 259 – 263: We have revised the sentences. As mentioned in line 155 – 170, our present results of *in vivo* expression analysis of *PTCH2*, *PTHrP*, *COL2* and *COL10* demonstrated that at least chondrocyte proliferation and subsequent differentiation to hypertrophic chondrocytes is also inhibited in the *Cp/Cp* embryos due to the loss of IHH signaling in the *Cp/Cp* embryos but delayed in the *Cp/+* embryos because of the lower amount of IHH.

- A schematic comparison of the known NHEJ pathway similarities/differences between humans, mice and chickens would be helpful.

This difference between birds and mammals has been unknown because there has been only one report on the chicken DSB repair pathway, which used the DT40 cell line. The HR pathway considered to be more important for the DSB repair in chicken DT40 cells than in mouse ES cells, suggesting that the roles of the two DSB repair pathways appear to be somewhat different between two species [Takata et al., *EMBO J.* 17, 4497-5508, 1998]. Further studies are needed to answer the reviewer’s comment.

- Lines 285-286 claim the Cp model “highlights the importance of NHEJ function I normal neurogenesis”. The production of neurones has not been assessed (even if reduced, apoptosis of the neuroepithelium before the onset of neurogenesis would be likely to limit neuron production). This conclusion should be removed.

Line 336: The sentence has been removed following the suggestion.

Reviewer #2 (Remarks to the Author):

Major

1) From this work, the conclusion that NHEJ1 deficiency “alone” causes embryonic lethality in the Creeper chicken can be suggested but not directly made. Strict demonstration would require a chicken with only the NHEJ1 defect. Therefore the title of the manuscript should be “Combined deletions of IHH and NHEJ.

Title: “Combined” has been added at the beginning of the title.

Minor

1) P 5 line 95. NHEF1.....NHEJ1

Line 101: “NHFJ1” has been corrected to “NHEJ1”

2) Would it be possible to perform clonogenic survival assays after DSB induction (IR or radiomimetic drugs) with the established primary cells to further support the NHEJ defect.

The growth of the *Cp/Cp* fibroblast cells were very slow in culture, and the colony forming cell assay was very difficult due to the low ability of proliferation. So we have not done this assay.

3) P 10 line 226 discovery of NHEJ1....it is a co-discovery by ref 27 and 28

Line 271 – 272: Following the suggestion, we have rewritten the sentence as follows (underlined). “Human NHEJ1, also known as Cernunnos or XLF, encoded by the *NHEJ1* gene was discovered as an XRCC4-interacting protein by a yeast two-hybrid screening system and as the protein mutated in patients with growth retardation, microcephaly, and immunodeficiency.³⁷.

38 ”

4) P 10 Lines 236-237. The CTR of NHEJ1 contains the NLS but also the Ku binding motif. Recruitment and retention of XLF at DSBs in cells requires Ku interaction see and refer to the work of Charbonnier, Caldecott and Chen. The data in this manuscript clearly show that truncated XLF is not recruited to sites of damage but that it is still can be localized in the nucleus.

Line 294 – 300: Following the suggestion by the Reviewer, we have added new sentences in the Discussion section citing three papers as follows. “The CTR of NHEJ1 contains the nuclear localization signal (NLS) but also the Ku binding motif, and recruitment and retention of NHEJ1

at DSBs in cells requires Ku interaction (Yano et al., 2008; Grundy et al., 2016; Nemoz et al., 2018). The present results clearly showed that the truncated NHEJ1 is not recruited to sites of damages but that it is still can be localized in the nucleus. Altogether, our findings strongly support that the Ku binding motif in the CTR of NHEJ1 is important for recruitment and retention of XLF at DSBs.”

5) P 12 line 281. Remove “was”

Line 332: “was” has been removed.

Reviewer #3 (Remarks to the Author):

Main concerns:

1. The morphogenetic abnormalities of the early stages of Creeper embryos (title of the first part of the results) are not well characterized. That is an important piece of result that should be in the main figures. The only information are in Suppl Fig 3G where we can see a Tunnel staining in a mutant but not control embryos and a descriptive table: hypoplasia of heart and brain and abnormal island formation should be better documented (mutants versus control embryos).

We have showed the photographs that clearly demonstrate morphological abnormalities of the *Cp/Cp* embryos at E3 in Fig. 1b. We have added more data on morphologies of *+/+*, *Cp/+*, and *Cp/Cp* embryos in Supplementary Tables 2 and Table 3.

2. The description of the deletions are very informative and important to this manuscript, however to be sure that the same deletions are causing the initial Creeper phenotype it will be informative to double check that these deletions exist in the JB and/or the Chinese strains too.

Unfortunately we cannot obtain the sample of the Chinese Xingyi bantam chicken. However, the size and chromosomal location of the deletion in our *Cp* mutant (the JB and GSP/*Cp* strains) was completely different from that in the Chinese strain reported in Jin et al. (*Sci. Rep.* 6, 30172, 2016). We have tried amplifying DNA fragments of the mutation site of the *Cp/Cp* embryos in the JB and GSP/*Cp* strains with the primers used in Jin et al. (2016). As a result, no PCR products were obtained because their forward primer was designed in the 578-bp inverted region contained in the 25 kb deletion in our JB and GSP/*Cp* strains. This result indicates that at least the 25 kb deletion containing the fifth and sixth exons of *NHEJ1* and the whole region of *IHH* in the *Cp* allele of the JB and GSP/*Cp* strains is different from that of the Chinese strain.

Minor concerns:

1. There is no clear rationale (in the main) text for choosing the GSP strains to study the mutation.

Line 52 – 54, Line 74 – 77: The GSP strain is the highly inbred strain derived from the Fayoumi breed, which shows quite low heterozygosity; its average heterozygosity is less than 0.01 for more than 50 microsatellite DNA markers (Nunome et al. *Exp. Anim.* 68, 177–193, 2019). We constructed a congenic strain of the *Creaper* allele that was introduced from the JB strain to characterize the *Cp* phenotype in homozygous genetic background. We have added the part that is deficient.

2. There is no explanation on why ALAD gene is being investigated and not another gene.

Line 140 – 141: We studied expression pattern of ALAD in the extraembryonic blood vessels and blood island according to GEISYA web site (<http://geisha.arizona.edu/geisha/>). It was expressed in the blood island as in the web site. We used its expression as a marker of the blood island. We added this explanation in the result session.

3. A bit of introduction on DSB would be welcome for the readers.

L264 – 269: We have added the introduction of DSBs with some references.

4. The panels of the figures are very small and are difficult to see (in particular Fig. 3 and Fig. 4).

Fig. 3: The photographs have been magnified to make it easy to see the signals in Fig. 3a by changing the arrangement of figures.

Fig. 5: The arrangement of Fig. 5a and 5c has been modified to magnify them.

5. Sentence starting at line 205 (discussion) is confusing: the deletion is likely the same in the Chinese strain (see main concern 2). Are they potentially different genetic mutation causing the Creeper phenotype? This is unclear to me.

L113 – 115, L233 – 237: As mentioned above, the mutation in the JB strain was different from that in the Chinese Xingyi bantam strain because no genetic fragments could be amplified with the former PCR primer that was used for detecting the mutation in the Chinese chicken.

However, the mutation in other Japanese native chicken breeds with the Creeper phenotype (Miyaji-dori, Jitokko) was the same

REVIEWERS' COMMENTS:

Reviewer #1 (Remarks to the Author):

The authors have made substantial changes to the text and added new experimental data which address my previous comments.

Reviewer #3 (Remarks to the Author):

The authors have satisfactorily responded to all my concerns and made the necessary changes to the manuscript. In particular, they improved the characterisation of the phenotype and clarified the explanation of the mutations in other strains.